# Screening efficiency of the Mood and Feelings Questionnaire (MFQ) and Short Mood and Feelings Questionnaire (SMFQ) in Swedish help seeking outpatients

Håkan Jarbin[1☯¤]*, Tord Ivarsson[2☯], Markus Andersson[1☯], Hanna Bergman[3‡], Gudmundur Skarphedinsson[4‡]

1 Faculty of Medicine, Department of Clinical Sciences Lund, Lund University, Child and Adolescent Psychiatry, University of Lund, Lund, Sweden, 2 Department of Mental Health, Regional Centre for Child and Youth Mental Health and Child Welfare, Faculty of Medicine and Health Science, RKBU Midt-Norge, Norwegian University of Science and Technology (NTNU), Trondheim, Norway, 3 Child and adolescent psychiatry, Region Halland, Varberg, Sweden, 4 Faculty of Psychology, University of Iceland, Reykjavik, Iceland

☯ These authors contributed equally to this work.
¤ Current address: BUP kliniken, Psykiatri Halland, Länssjukhuset, Halmstad, Sweden
‡ These authors also contributed equally to this work.
* hakan.jarbin@regionhalland.se

**Data Availability Statement:** All relevant data are within the manuscript and its Supporting Information files.

## Abstract

### Background

To evaluate screening efficiency and suggest cut-offs for parent and child Mood and Feelings Questionnaire (MFQ) and the short version (SMFQ) in unselected help seeking child- and adolescent psychiatric outpatients for subgroups of 6–12 versus 13–17 year olds and boys versus girls.

### Method

Eligible for inclusion were newly admitted outpatients age 6–17 years (n = 5908) in four Swedish child- and adolescent psychiatry clinics. They were prompted consecutively and n = 307 accepted a specific day for assessment until time slots randomly were filled. We prospectively validated the MFQ (33 items) and SMFQ (13 items) in patients (n = 186) using receiver operating characteristics against a reference test of Longitudinal Expert All Data DSM-IV depression based on a Kiddie-Schedule for Affective Disorders and Schizophrenia and 1.2 (sd .6) years of follow-up.

### Results

A depressive disorder was confirmed in 59 (31.7%) patients ranging from 14.0% for girls 6–12 years to 53.3% for girls 13–17 years. SMFQ performed roughly equivalent to MFQ. Adolescent score on SMFQ discriminated fairly for boys with Area Under Curve .77 (95% confidence interval .59–.81) and good (.82, .69–.91) for girls and parent ratings for adolescent girls (.85, .73–.93), but not for boys. Depression in children below age 13 could not be

**Funding:** HJ, 2008-22893, Stiftelsen Söderström-Königska, https://www.sls.se/vetenskap/sok-anslag/stift.-soderstrom/, NO role in design, data, publishing, preparation MA, 110361, Development and Education (FoUU) within Region Halland, Sweden, https://vardgivare.regionhalland.se/utveckling-forskning/forskning/projektmedel-och-bidrag-for-forskning-och-utveckling/doktorandmedel/, NO role in design, data, publishing, preparation HJ, 133821, Development and Education (FoU) within Region Skåne, Sweden, https://www.skane.se/organisation-politik/forskning/sa-finansierar-vi-forskningen/, NO role in design, data, publishing, preparation.

**Competing interests:** The authors have declared that no competing interests exist.

discriminated by MFQ or SMFQ whether filled in by child and mostly also when filled in by parent. Favouring maximum kappa value, the optimal cut-off was for MFQ self-report girls $\geq$32 versus boys $\geq$11 and for SMFQ self-report girls $\geq$17 versus boys $\geq$ 6. Suggested clinical SMFQ cut-offs for girls were $\geq$12 and for boys $\geq$ 6.

## Conclusions

MFQ and SMFQ can, with gender-based cut-offs, be used for screening in clinical populations of adolescents but not in children. Parent MFQ and SMFQ can be used for adolescent girls but not boys. SMFQ is sufficient for screening.

## Introduction

Depression is a common and increasing burden on adolescents [1, 2]. Depression affects children in about half [3] to one third [2] of the rate of adolescents and does not seem to increase before puberty [2]. Depression confers a range of risks on education, relationships, suicidality, abuse and on physical health [4–7] In spite of these compelling data, there is a disturbing under-recognition of depression and more so in adolescents than in adults [8] while data on recognition in children are lacking. Most adolescents with depression do not receive treatment [1]. Our Swedish clinical outpatient sample indicated a prevalence of about 30% with depression [9] in sharp contrast to the 10% prevalence of clinical depression diagnoses among all new referrals to five large Swedish outpatient clinics in 2017 (data on file). These troubling findings highlight the urgent need for better tools for detecting depression in a clinical context. Questionnaires to patient or parents can be an efficient way to improve detection of depression in clinical populations.

The Mood and Feelings Questionnaire (MFQ) was developed to screen for depression in epidemiological samples [10]. It has also been validated in clinical samples [11–16] and is recommended as a screening tool [5]. The largest clinical validation contained ongoing patients from ADHD and mood disorder clinics along with high risk and epidemiological samples in the U.S. MFQ performed well for child and parent ratings and in both genders and in children as well as in adolescents with area under curve (AUC) mostly good (.80–.90). However, the MFQ child report did somewhat less well in clinical patients with externalizing disorders [11]. Parent MFQs have discriminated as well as child ratings in most [11, 12, 14] but not in all [16] comparisons. However, the parent MFQ did worse when the diagnosis was established with just an interview with the child [16] as opposed to studies which used separate interviews with child and parents [11, 12] or a clinical diagnosis [14]. On the contrary, MFQ-parent was better than MFQ-child at discriminating any mood disorder. A combination of parent and self-ratings was better than either rating alone. Further, a lower AUC to discriminate mood disorder was noted in MFQ-parent from clinical as opposed to non-clinical studies [11]. Earlier studies have suggested cut-offs based on maximum separation of 27–29 (MFQ-child) and 21–27 (MFQ-parent) [11–13, 16]. Suggested cut-offs favouring sensitivity were 26–28 (MFQ-child) and 22 (MFQ-parent) [14, 15].

The Short Mood and Feelings Questionnaire (SMFQ) was developed to enhance clinical and epidemiological use [17]. Validation studies of the SMFQ child versions have shown significant discrimination of depression but still quite divergent AUCs with two studies reporting fair AUCs of .72–.73 [17, 18] while other reported good AUCs of .84–.87 [13, 15, 19, 20]. Lower discriminations were seen with 12 year olds [18] and ages 11–16 years in an

epidemiological sample [17] while the better discriminating studies were mostly done with older samples i.e. asthma patients and controls with mean age 14 years [19] detained 15–16 year olds [13], adolescents mean age 15.6 years recruited for CBT in depression [15] or 18 year olds in an epidemiological sample [20]. Suggested cut-offs on the SMFQ self-report have been divergent with cut-offs ranging from 4–5 in studies with just fair AUC and younger subjects [17, 18] to a high of 10–12 in studies with good AUCs and older subjects. Two studies have also compared the SMFQ and MFQ patient versions and reported high correlations and identical AUCs [13, 15]. Three studies have compared the discriminating abilities of parent versus child SMFQ with somewhat opposing results as the child version did better in the younger group of 6–11 year olds [10], the parent version did better in 11–15 year olds [17] while they were equal in 13 year olds [18].

A clinical Swedish study of the ten-item Montgomery Åsberg Depression Rating Scale-Self report versus a semi structured Schedule for Affective Disorders and Schizophrenia for School-Age Children (K-SADS) clinician rated diagnosis of depression revealed significant and intriguing gender differences with girls needing a higher cut-off score than boys (16 versus 11) for maximum kappa separation in adolescents with mean age 15 years [21] and contrary to the suggested equal cut-off scores for SMFQ in adolescent boys and girls [15]. However, the MADRS-S study used a clinician rated K-SADS interview as criterion in the study while the SMFQ study just used an arbitrary cut-off on the self-reported child depression rating scale-revised (CDRS-S) as criterion in that study. MFQ studies have not reported analyses of different cut-offs from boys and girls or from children and adolescents.

To sum up, MFQ-child and MFQ-parent has repeatedly been shown to discriminate a depressive disorder most often in the good range but somewhat worse in a clinical setting or for MFQ-child with externalizing disorders. SMFQ-child and SMFQ-parent has correlated strongly to MFQ and has discriminated as MFQ in adolescents but just fair in younger samples. MFQ and SMFQ have usually been validated in epidemiological, high risk or among already diagnosed and clinical samples but not in unselected help seeking child psychiatric outpatients except the MFQ Arabic version in a small study of mostly adolescents [14]. Another recent study included help seeking adolescents but they were recruited and selectively referred for a cognitive behaviour intervention for depression [15]. There is a need to evaluate MFQ and SMFQ in unselected samples of help seeking child and adolescent psychiatry (CAP) outpatients before any other diagnostic activities have been accomplished, i.e. in the proper context for a clinical screening measure.

### Aims

The aims were: a/ to assess criterion validity of MFQ and SMFQ from self and parent ratings compared to Longitudinal Expert All Data (LEAD) depression diagnoses as the criterion measure in an unselected clinical sample with comparisons for gender and age and b/ to arrive at suggested cut-offs for use in clinical samples.

## Materials and methods

### Participants

There were 307 patients assessed in a diagnostic study from a consecutive sample of 5908 out-patients of ages 6–17 years, who sought treatment at four CAP clinics in Sweden from January 2010 to March 2013. All new referrals were routinely interviewed by phone and 5553 (94%) consented to participate in a diagnostic study. The parents, who had given consent, were then in a consecutive manner asked to participate at the available time slots. If the suggested date was not suitable, the next consecutive patient at the site was asked. In order to arrive at roughly

equal numbers between children and adolescents and between boys and girls, about 15 (6%) children were at later stages of recruitment actively selected in order to include more pre-pubertal girls. Forty cases were discarded due to protocol violations in the K-SADS as all assessments from two clinicians were incomplete. The included sample was, compared to the entire outpatient sample, more affected by internalizing symptoms (Cohen's d about 0.3) and externalizing symptoms (Cohen's d about 0.5) as assessed by parental report. A more detailed description of the sample and recruitment procedures can be found at previous publications [9]. Further, MFQ data were missing from 35 parents due to administrative failure at one of the centres. Thus, parents of 232 patients filled in MFQ. Mothers' MFQ data were used as parental MFQ in 211 (90.9%) cases and father ratings, when only father ratings were available, were used in 21 (9.1%) cases. In another 36 cases, patients declined to fill in the MFQ. Data from the remaining n = 186 cases with both parental and child MFQ are reported (see Fig 1). Mean age was 12.7 (sd. 2.9, range 6.1–17.8) years, without significant (p = 0.15) gender difference, boys 11.9 (sd 2.9, range 6.2–17.7) versus girls 12.5 (sd 3.4, range 6.1–17.8) years. The proportion of children 6–12 years was n = 101 (54.3%). There were more boys (n = 102, 54.8%) than girls. The sample studied was compared to a concurrent and consecutive sample (n = 5641) of help-seeking CAP outpatients [22]. The sample was no different in proportions of children versus adolescents or boys versus girls but slightly more symptomatic in internalizing ($d \approx 0.3$, p< 0.001) and in externalizing symptoms ($d \approx 0.5$, p< 0.001).

## Procedures

Parents of new referrals to outpatient CAP clinics were interviewed in a standard intake procedure by phone for a baseline assessment and triaging. At the end of the phone interview, the parent was informed about the diagnostic study. Within six weeks the participants, who had given oral consent, were offered a day of assessments including the MFQ, various other scales and the semi-structured K-SADS- Present and Lifetime version (K-SADS-PL) until time slots randomly were filled. Patients and parents were interviewed separately. Adolescents started with K-SADS-PL while parents were interviewed afterwards while adolescents filled in the MFQ. Children, on the other hand, started with the MFQ as, per suggestion, parents are to be interviewed before prepubertal children in the K-SADS-PL. The clinician performing the K-SADS-PL and subsequent clinicians were blind to the MFQ. The K-SADS-PL interviews yielded clinician established DSM-IV diagnoses, which were added to the clinical records. Later on, "Longitudinal Expert All Data Diagnoses" (LEAD) diagnoses were arrived at by two senior consultants, who reviewed the K-SADS-PL and all subsequent clinical information after a mean follow-up of 1.2 (sd. 0.6, range 0.1–3.1) years [9].

## Ethics

The Ethical Review Board at Lund University approved the study. Patients aged 15 years and above and parents consented to the study in writing.

## Measures

**Longitudinal Expert All Data (LEAD) diagnoses.** LEAD diagnoses were based on the K-SADS-PL 2009 version and further clinical work-up and follow-up. A blinded reliability test of 30 cases of the LEAD procedure was performed between the two senior authors (HJ and TI) using randomly chosen cases. Kappa values for spectrum diagnoses were excellent with ADHD, behaviour, and anxiety disorders in full agreement and for depressive disorders at kappa 0.92. A comprehensive description of measures and procedures can be found in a previous report on the KSADS using the same sample [9]. Subsequently, the clinical records were

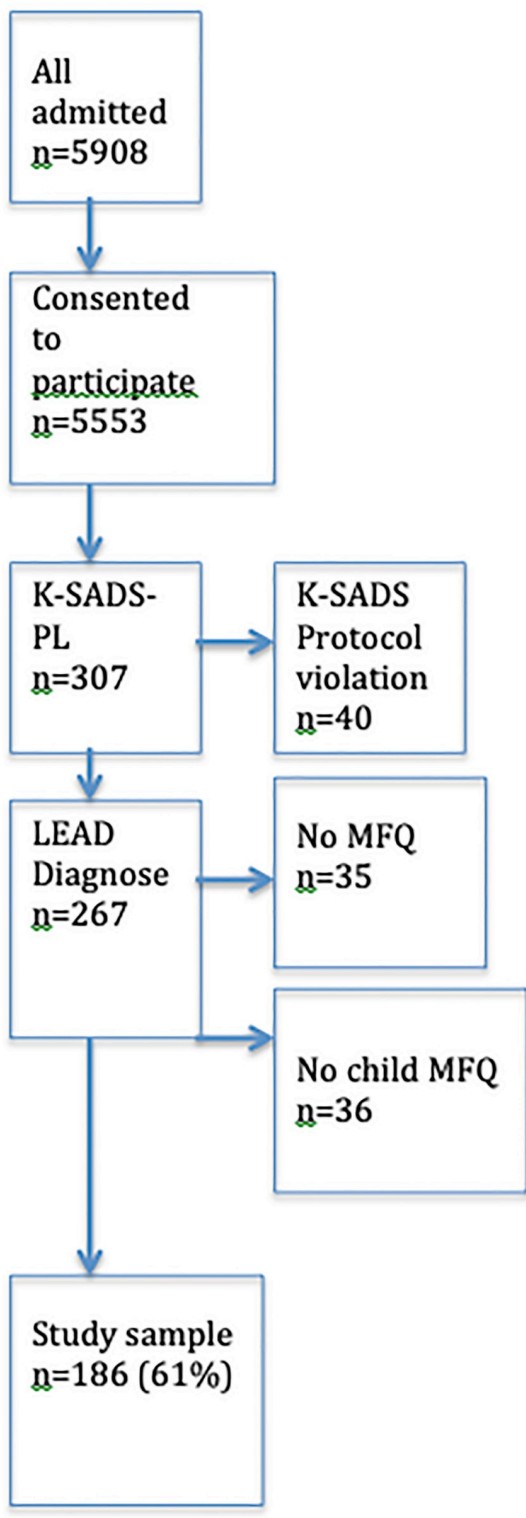

**Fig 1. Flow chart of recruitment.**

further evaluated using a LEAD process [9] commonly viewed as the "gold standard" to evaluate semi-structured interviews, which needs to consider all information brought in through diagnostic procedures, the level of impairment and the outcome of treatment across a suitable time period [23–25].

**The Mood and Feelings Questionnaire (MFQ).** The MFQ is a questionnaire assesses depressive symptoms over the previous two weeks. Responses are rated on a three-point scale (0 = not true, 1 = sometimes true, and 2 = true). Total score ranges from 0–68 with a suggested cut-off usually ≥29. MFQ was developed to assess depressive symptoms in youth aged 8–18 years. There are matching versions for self (with 33 items) and parent report (with 34 items) [10] and a short MFQ with 13 items [17]. Internal consistency has been high with Cronbach's alphas for MFQ about .90–.96 [10, 11, 13–15, 26] and for SMFQ about .85–.89 [10, 13, 15, 19]. Correlations between the long and short versions have been very strong (r = .95) [13]. Correlations between MFQ-parent and MFQ-child have been fairly strong (r = .51–.53) [12, 14, 16].

## Statistics

A Chi-square test was used to analyse gender differences in prevalence of disorders. Internal consistency reliability of the MFQ and SMFQ was assessed with the Cronbach´s alpha, which is a measure of correlations between the items in a scale. Alpha 0.90 or above is considered excellent and .85–.89 as good [27]. T-tests were conducted to evaluate gender differences in prevalence of disorders and to assess diagnostic group differences (depressive disorder versus non-depressive disorder) on the MFQ and SMFQ filled out by patients and parents. Pearson's bivariate correlation was conducted to assess the correlation between parent and patient ratings on the MFQ. Receiver operating characteristics (ROC) analyses were conducted to examine the screening efficiency of MFQ and SMFQ child and parent report (index tests) versus the reference standard of a LEAD diagnosis of any DSM-IV depressive disorder as criterion measure. Generally, the area under the curve (AUC) in a ROC curve is at .50 no better than chance and at 1.0 a perfect diagnostic accuracy. Cut-offs for interpreting the AUC values are often categorized in the following way: 0.6 to 0.7 (poor), 0.7 to 0.8 (fair), 0.8 to 0.9 (good) and above 0.9 (excellent) [28]. Agreement between cut-off scores for MFQ or SMFQ and depression were also evaluated using the Kappa statistic. Cut-offs for interpreting kappa values are frequently categorized as follow: 0 to 0.20 (poor), 0.21 to 0.40 (fair), 0.41 to 0.60 (moderate), 0.61 to 0.80 (good) and above 0.80 (very good agreement) [29]. Optimal cut-offs were chosen to minimize false-positives and false-negatives equally and focused on maximizing efficiency [30] but cut-offs for prioritizing sensitivity or specificity at .80 are also displayed in Table 3. These cut-off points are presented with their sensitivity to detect depression, which is the proportion of depressions that are correctly identified by the scale. The cut-offs are also presented with their specificity, which is the proportion of non-depression correctly identified as non-depression [31]. Choosing a cut-off in clinical practice needs to address the purpose of the screen. Usually, depression detection sensitivity should be prioritized to make sure that detection is adequate. We choose cut-offs for suggested clinical use with sensitivity of about .75 or better if sensitivity was too low with the cut-off for best separation.

Tests were two-tailed. Alpha was set to .05. SPSS version 24 was used for analyses and MedCalc statistical software for ROC analyses. Data are available in a supplementary file.

## Results

### Sample characteristics

Depressive disorders were diagnosed in 59 (31.7%) with major depression in 35 (18.8%), dysthymia in 7 (3.8%) and depression not otherwise specified in 17 (9.1%) patients. In the four

subgroups, depression was diagnosed in 8 of 57 (14.0%) girls 6–12 years, in 19 of 97 (19.6%) boys 6–12 years, in 21 of 53 (39.6%) boys 13–17 and in 32 of 60 (53.3%) girls 13–17 years old. Further, attention-deficit/hyperactivity disorder (ADHD) was diagnosed in 114 (61.3%), anxiety disorder (generalized anxiety -, separation anxiety—or social anxiety disorder) in 64 (34.4%), disruptive disorder in 60 (32.3%) and autism spectrum disorder in 20 (10.8%).

Gender difference across disorders was seen for ADHD (boys 72.5% versus girls 47.6%; $\chi^2$ = 12.1, p = .001) but not for depression or other disorders. See also S1 Table.

## Internal reliability

MFQ showed excellent ($\alpha$ = 0.93) and SMFQ good ($\alpha$ = 0.87) Cronbach´s alpha for parents (n = 232) and patients (n = 186).

## MFQ and SMFQ total score across age, diagnosis and gender

MFQ scores for girls were higher than for parents (p< 0.001) and ratings between girls and their parents correlated (r = .44, p < 0.001), and most so for adolescent girls (r = 0.51, p< 0.001). The MFQ score levels for boys and their parents did not differ (p = 0.115) but did not significantly correlate (r = 0.19, p = 0.056)(Table 1).

Patients with depression had significantly higher scores than non-depressed for both child and parent ratings and with both MFQ and SMFQ (S2 Table).

There were marked age and gender differences in MFQ scorings. Teen-girls scored the highest while preadolescent boys scored the lowest. Differences were not as marked in parent ratings, but parents scored higher for adolescents in both MFQ and SMFQ (S3 Table).

## Screening efficiency

Adolescent's score on MFQ and SMFQ predicted a diagnosis of depressive disorder with fair to good AUC (.77–.82) and moderate kappa (.46–.60). Parent score on MFQ and SMFQ predicted a diagnosis of depressive disorder in adolescent girls only with good AUC (.85) and a good kappa agreement (.60–.63) but not for adolescent boys. Sensitivity when using cut-offs for maximum kappa for adolescent depression ranged from 64% for girls to 100% for boys while specificity ranged from 57% for boys to 96% for girls.

**Table 1. MFQ means, standard deviations and independent t-test as per child versus parent ratings separately for boys and girls and for children and adolescents.**

| subgroup | Patient M (SD) | Parent M (SD) | R | t-testF | p-value |
|---|---|---|---|---|---|
| Pre-boys[a] (n = 67) | 14.5 (10.7) | 12.3 (8.6) | .10 | 1.336 | .186 |
| Pre-girls[a] (n = 34) | 20.5 (13.9) | 12.4 (10.6) | .24** | 3.075 | .004 |
| Teen-boys[b] (n = 35) | 19.8 (15.6) | 17.0 (13.4) | .20 | 0.901 | .374 |
| Teen-girls[b] (n = 50) | 29.4 (14.6) | 16.7 (13.4) | .51*** | 6.403 | < .001 |
| All boys (n = 84) | 16.3 (12.8) | 13.9 (10.7) | .19 | 1.589 | .115 |
| All girls (n = 102) | 25.8 (14.9) | 15.0 (12.5) | .44*** | 6.775 | < .001 |
| All (N = 186) | 20.6 (14.5) | 14.4 (11.5) | .32*** | 5.472 | < .001 |

Correlations describe agreement between child and parent.

[a] Pre-Boys/Girls means 6–12 years old,

[b] Teen-Boys/Girls means 13–17 years old.

** < .01,

*** < .001

**Table 2. Psychometric properties of the Mood and Feelings Questionnaire (MFQ) and Short Mood and Feelings Questionnaire (SMFQ) for the child and parent ratings versus the criterion test with Longitudinal Expert All Data diagnoses of DSM-IV depressions.**

| s | MFQ scale | AUC[c] | CI[d]95% | P | Cutoff | Sens[e]% | Spec[f]% | Kappa |
|---|---|---|---|---|---|---|---|---|
| **Pre-girls[a]** | Child MFQ (n = 34) | .52 | .34–.70 | .872 | 21 | 17 | 82 | .06 |
| | Child SMFQ (n = 34) | .51 | .33–.68 | .950 | 7 | 17 | 96 | .18 |
| | Parent MFQ (n = 52) | .70 | .55–.82 | .141 | 16 | 67 | 80 | .31 |
| | Parent SMFQ (n = 52) | .73 | .59–.84 | .086 | 8 | 67 | 83 | .34 |
| **Pre-boys[a]** | Child MFQ (n = 67) | .53 | .41–.65 | .762 | 26 | 36 | 91 | .29 |
| | Child SMFQ (n = 67) | .53 | .40–.65 | .769 | 14 | 29 | 92 | .25 |
| | Parent MFQ (n = 82) | .70 | .59–.80 | .009 | 23 | 31 | 94 | .30 |
| | Parent SMFQ (n = 82) | .62 | .51–.73 | .094 | 7 | 63 | 65 | .20 |
| **Teen-girls[b]** | Child MFQ (n = 50) | .81 | .67–.91 | < .001 | 32 | 72 | 88 | .60 |
| | Child SMFQ (n = 50) | .82 | .69–.91 | < .001 | 17 | 64 | 96 | .60 |
| | Parent MFQ (n = 55) | .85 | .73–.93 | < .001 | 12 | 83 | 76 | .60 |
| | Parent SMFQ (n = 55) | .85 | .73–.93 | < .001 | 4 | 87 | 76 | .63 |
| **Teen-boys[b]** | Child MFQ (n = 35) | .78 | .61–.90 | < .001 | 11 | 100 | 57 | .52 |
| | Child SMFQ (n = 35) | .77 | .59–.89 | .001 | 6 | 93 | 57 | .46 |
| | Parent MFQ (n = 43) | .63 | .47–.78 | .134 | 11 | 76 | 58 | .32 |
| | Parent SMFQ (n = 43) | .64 | .48–.78 | .118 | 6 | 76 | 50 | .24 |

[a] Pre-Boys/Girls means 6–12 years old,

[b] Teen-Boys/Girls means 13–17 years old,

[c] Area Under Curve,

[d] Confidence Interval,

[e] sensitivity,

[f] specificity

Depression in children younger than 13 years could not be predicted by MFQ or SMFQ except for parents predicting depression with a fair accuracy (.70) and a fair kappa (.30) in preadolescent boys (Table 2).

Further analyses are displayed for adolescents only and with just SMFQ as screening efficiencies for 6–12 year olds were poor and in adolescents roughly equal between MFQ and SMFQ. Optimal cut-offs to minimize false-positives and false-negatives equally and focusing on maximizing efficiency but also cut-offs for prioritizing sensitivity or specificity at .80 are displayed in Table 3. Optimal cut-off varied with a low of 4 in parent rating for adolescent girls to a high of 17 for adolescent girls self-rating. Adolescent boys self-rating showed an optimal cut-off at a score of 6 with sensitivity.93 and specificity of .57. Adolescent girls self-rating showed an optimal cut-off at a score of 17 with sensitivity .64 and specificity .96. Parent ratings for adolescent girls showed an optimal cut-off at a score of 4 with sensitivity at .87 and specificity at .76.

Adjusting the suggested cut-offs for clinical use to arrive at a minimum of about .75 sensitivity resulted in no changes for parents of girls or teenage boys but a lowered cut-off for teenage girls on the SMFQ from 17 to 12. This cut-off conferred still a moderate kappa of .48.

## Discussion

MFQ and SMFQ were, in this highly comorbid and unselected clinical sample, unable to predict depression in prepubertal children, which is at odds with previous studies. For adolescents, both the MFQ and SMFQ were adequate. We recommend SMFQ for screening in

**Table 3. Psychometric properties at cut-offs to either prioritize sensitivity, specificity or best separation on the Short Mood and Feelings Questionnaire (SMFQ) for the prediction of depression in adolescents.**

| SMFQ scale | Cut score | Sens[b] | Spec[c] | Kappa |
|---|---|---|---|---|
| Teen-boys[a] self rating | 80% sensitivity and Max Kappa at score ≥6 | 93 | 57 | 0.46 |
| | 80% specificity at score ≥10 | 50 | 86 | 0.38 |
| Teen-girls[a] self rating | 80% sensitivity at score ≥9 | 80 | 64 | 0.44 |
| | Suggested cut off at score ≥12 | 72 | 76 | 0.48 |
| | Max Kappa at score ≥17 | 64 | 96 | 0.60 |
| | 80% specificity at score ≥13 | 68 | 80 | 0.48 |
| Teen girls[a] parent rating | 80% sensitivity at score ≥7 | 80 | 76 | 0.56 |
| | Max Kappa at score ≥4 | 87 | 76 | 0.63 |
| | 80% specificity at score ≥9 | 63 | 88 | 0.50 |

[a] Teen-Boys/Girls means 13–17 years old.

[b] sensitivity,

[c] specificity.

clinical populations due to brevity but we suggest gender-based cut-offs. Adding a parental SMFQ improves the prediction for girls but not for boys.

The sample was highly comorbid and in that way representative of referrals for specialized CAP services. Previous studies have examined patients already diagnosed and often selected from subspecialized units [10–12, 16, 17], where they have probably received psychoeducation including explanations of signs and symptoms. This will likely lower the "noise" of comorbidity and increase patient and parent mutual awareness of symptoms. Other studies have examined MFQ in population-based samples [20, 26, 32, 33], where rates of comorbidity and severity are considerably lower than in referred samples. In the Lebanese study of MFQ in unselected referrals [14], patients, were in comparison to our sample, more often female (64 versus 45%, p < .01), slightly older (13.9 versus 12.7 years old) and with less ADHD (21.7 versus 61.3%, p < .001) underlining that help seeking CAP patients in an Arabic versus in a Scandinavian society differs. Thus, our data are valid when using MFQ as a screening measure in unselected referred patients in Scandinavia and similar western cultures.

Levels on MFQ also illustrate disparities in sampling. Our levels on MFQ-patient for depressed parallel those of the diverse American clinical and non-clinical sample of about the same mean age of 13 years [11] while the non-depressed in our sample had higher scores and thus separated less from depressed cases. Studies with a female preponderance [14, 16] or from a clinic for depression [12] have shown higher levels of MFQ-patient and a larger difference from non-depressed than ours. In MFQ-parents our data on levels for depressed and non-depressed patients are lower than in samples with more girls [14, 16] or higher rates of depression [12] but are also lower than in the large American sample [11] and generally differ to a smaller degree from the non-depressed in other samples. Level of SMFQ is seldom reported and only figures for a sample of 6–11 year olds [10], which, as expected, had slightly lower levels on both self and parent report than our sample of 6–17 year olds. Our large number of untreated ADHD and disruptive disorder can explain the smaller difference between depressed and non-depressed in our sample as patients with externalizing disorders can be expected to experience symptoms of depression due to their load of stress and conflict. However, this is the clinical environment for unselected outpatients and screening tools of today.

Subgrouping of MFQ-patient levels across age and gender have been reported with girls scoring higher than boys but no difference across age [26, 34] while our clinical sample as expected had higher levels but also an age difference with adolescents scoring higher than

children. MFQ-parent has not been reported across age and gender in previous studies. In our sample, parents scored higher for their adolescent child without gender difference, probably reflecting the larger proportion of depression in adolescents in our clinical sample.

A very surprising and intriguing finding is the lack of correlation between boys and their parents' ratings on MFQ, which is at odds with earlier reports [12, 14, 16]. This was obvious for both preadolescent and teenage boys. Contrarily, MFQ scores from our girls and their parents correlated significantly, as expected and in line with other studies. This can possibly be explained by using MFQ in an early stage of the diagnostic process as opposed to samples when patients already had received a formal diagnosis. The Arabic unselected and help seeking study also showing a significant child-parent correlation recruited a larger proportion of girls and somewhat older subjects [14]. These factors increased correlation in our sample. A possible explanation is that girls and most so adolescent girls have communicated to parents on their feelings and thoughts before first evaluation while boys have not. If MFQ would be administered at a later stage after diagnosis and treatment, the correlation between boys and their parents might have increased and become significant. However, an ecologically valid test of a screening measure should be carried out in the very same condition as the recommended use and that is not to test a screen on already diagnosed patients but rather before any diagnostic procedure. Thus, the surprising non-correlation between boys and their parents might add to the ecological fit for our data. Further, it suggests that in clinical practice adolescent boys needs to be screened and interviewed separately as they might have significant symptoms of depression unknown to their parents. A conjoint interview could restrict the rapport from the boy but maybe not from the girl as the parent already seem to know about symptoms of depression for girls.

The MFQ and SMFQ patient versions performed in our study rather well as a screening measure for adolescents albeit slightly below the AUCs from most [11, 13, 15, 16, 19, 20] but not [12, 14, 17, 18] all studies. Very surprising, the MFQ and the SMFQ patient versions were unable to detect depression in preadolescents completely at odds with previous studies including a study, which also analysed children separately [11] or in a sample from a depression clinic with a female preponderance and a high prevalence of depression [16]. This cannot be explained by a type II error of low numbers as our AUCs for children were very close to 0.50 i.e. not better than by chance. The most likely explanation is our very high prevalence of ADHD and externalizing disorders. Subjects with this comorbidity had just a fair AUC [11] but the screening properties for MFQ-patient in children with a high degree of ADHD and externalizing symptoms have not been reported separately. Children with a high load of externalizing symptoms might experience low mood and negative thoughts without having a depressive disorder. This is in line with the more undifferentiated presentations at large in younger children [35]. We suggest that MFQ and SMFQ in clinical screening for depression in unselected help seeking patients are used in adolescents but not in children.

Parental MFQ and SMFQ had largely poor or barely fair agreements for preadolescents except for in adolescent girls, were the AUC was good. Again, this is at odds with previous studies where parent MFQ generally had fair [14, 16, 18] or good [11, 12, 17] AUCs. Parental AUCs were surprisingly poor for adolescent boys. The most likely explanation is the high comorbidity in our sample and that parents often were unaware of depressive symptoms in their adolescent boy. We suggest that parental version is used to parents of adolescent girls but not to parents of boys or preadolescent girls.

Our statistically optimized cut-offs varied extremely between boys and girls at odds with previous studies who suggest gender-neutral cut-offs but in line with the Swedish MADRS-S study [21]. Teenage girls' cut-off score for optimized separation was more than double that of their parents or teenage boys. Two studies have reported AUC separately for boys and girls.

AUC was just slightly lower for boys but still good to fair in separating any mood disorder or major depression from non-mood versus a KSADS diagnosis [11] while boys performed largely just as good as girls but reference test was just a cut-off on a depression rating scale [15]. In our unselected help seeking sample, the suggested cut-offs from the point of best separation exhibited almost three fold levels for girls, which is surprising and also more than expected from the MADRS-S study. However, that study did not parse out children, which might have moderated girls' cut-off as pre-adolescent girls were included and they scored lower levels on MFQ more in line with boys in our sample.

For clinical purposes, we suggest using the SMFQ for adolescents with a cut-off yielding 80% sensitivity. Our adolescent girls had an optimized cut-off of 17, which gave a somewhat low sensitivity of 64%. Raising sensitivity to 80% resulted in a cut-off of 9 e.g. almost halved and severely compromised specificity and the kappa value. Unfortunately, a small sample size probably accounts for this dilemma. A reasonable cut-off is between 9 and 17 and somewhat in line with the most recent study of SMFQ in clinical adolescents [15] but opposed to older studies with mostly children and epidemiological samples. Our teenage boys arrived at both optimized cut-off and sensitivity above 80% with a cut-off of 6. Similarly, parents of teenage girls needed a cut-off of 7 for a sensitivity of 80% but optimal separation was at a cut-off of 4 yielding a sensitivity of 87%. Thus, in a non-selected sample of help seeking adolescents we suggest gender specific cut-offs with teenage girls at 12, boys at 6 and parents of teenage girls at 4.

## Strengths and limitations

The main strength of this study was the unselected sample of participants from a child and adolescent clinical population. All patients were new referrals without prior contact with psychiatric services. Thus, they had not received any prior psychiatric diagnosis, assessment or psychoeducation about depression. This recruitment is an ecologically suitable ground for testing the validity of a screening measure. The criterion or reference test of LEAD diagnoses were of good quality and independent from the MFQ scores as no information from the scales was included in the clinical records.

The main limitation is the sample size when analysing subgroups based on age and gender. Preadolescent girls and teenage boys had low numbers conferring a risk of type-II errors. The AUC of parent SMFQ in preadolescent girls was in the lower fair range but still did not reach significance, possibly due to low numbers. Secondly, adolescents filled in the MFQ after a semi-structured K-SADS-PL, which might have given them knowledge of the meaning of screening items while children were first given the MFQ without any preceding clinical interview whatsoever. This could possibly inflate adolescent agreement. Another limitation concerns generalizability as our help seeking population have high rates of ADHD and externalizing disorders, which might not hold true in other societies.

## Conclusions

MFQ and SMFQ can, with gender-based cut-offs, be used for screening in clinical populations of adolescents but not in children. Parent MFQ and SMFQ can be used for adolescent girls but not boys. SMFQ is sufficient for screening.

## Supporting information

**S1 Table. The frequency of psychiatric disorders in the outpatient sample with MFQ ratings from both parent and patient (n = 186).**
(DOCX)

**S2 Table. Means, standard deviations and independent t-test for depression or non-depression for child and parent ratings on the MFQ and SMFQ.**
(DOCX)

**S3 Table. Means, standard deviations, analysis of variance and independent t-test for MFQ and SMFQ with child and parent ratings separately for boys and girls and for children and adolescents.**
(DOCX)

**S4 Table. Checklist of StarD reporting guidelines.**
(DOCX)

**S1 Dataset.**
(SAV)

## Acknowledgments

The authors are grateful to families and the dedicated research team in the study.

## Author Contributions

**Conceptualization:** Håkan Jarbin, Tord Ivarsson, Markus Andersson.

**Data curation:** Håkan Jarbin, Markus Andersson.

**Formal analysis:** Håkan Jarbin, Tord Ivarsson, Hanna Bergman, Gudmundur Skarphedinsson.

**Funding acquisition:** Håkan Jarbin, Markus Andersson.

**Investigation:** Håkan Jarbin, Markus Andersson, Hanna Bergman.

**Methodology:** Håkan Jarbin, Tord Ivarsson.

**Project administration:** Håkan Jarbin, Markus Andersson.

**Software:** Gudmundur Skarphedinsson.

**Supervision:** Gudmundur Skarphedinsson.

**Validation:** Håkan Jarbin, Tord Ivarsson.

**Writing – original draft:** Håkan Jarbin.

**Writing – review & editing:** Tord Ivarsson, Markus Andersson, Hanna Bergman, Gudmundur Skarphedinsson.

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
