## [Decision Letter · Decision Letter 0]

7 Feb 2020

PONE-D-19-26460

Screening efficiency of the Mood and Feelings Questionnaire (MFQ) and Short Mood and Feelings Questionnaire (SMFQ) in Swedish help seeking outpatients

PLOS ONE

Dear Dr. Jarbin,

Thank you for submitting your manuscript to PLOS ONE. After careful consideration, we feel that it has merit but does not fully meet PLOS ONE’s publication criteria as it currently stands. Therefore, we invite you to submit a revised version of the manuscript that addresses the points raised during the review process.

We would appreciate receiving your revised manuscript by Mar 23 2020 11:59PM. To enhance the reproducibility of your results, we recommend that if applicable you deposit your laboratory protocols in protocols.io, where a protocol can be assigned its own identifier (DOI) such that it can be cited independently in the future. For instructions see: http://journals.plos.org/plosone/s/submission-guidelines#loc-laboratory-protocols

We look forward to receiving your revised manuscript.

Kind regards,

Ali Montazeri

Academic Editor

PLOS ONE

Journal Requirements:

3. Please ensure that you refer to Figure xxxxx in your text as, if accepted, production will need this reference to link the reader to the figure.

Reviewers' comments:

Reviewer's Responses to Questions

**Comments to the Author**

1. Is the manuscript technically sound, and do the data support the conclusions?

Reviewer #1: Yes

Reviewer #2: Yes

2. Has the statistical analysis been performed appropriately and rigorously? 

Reviewer #1: Yes

Reviewer #2: I Don't Know

3. Have the authors made all data underlying the findings in their manuscript fully available?

Reviewer #1: Yes

Reviewer #2: Yes

4. Is the manuscript presented in an intelligible fashion and written in standard English?

Reviewer #1: Yes

Reviewer #2: Yes

5. Review Comments to the Author

Reviewer #1: Dear Authors

Would you consider comments as follows:

Introduction

Since the introduction section describes both children and adolescents psychiatric disorders and instruments for detecting these problems, I’m wondering why authors began the section with depression in adolescents while the target group in the study was 6-17 years.

The section is not well organized and should be revised. For example, it is not clear what is the role of K-SADS here. .

The aim of the study should be mentioned as separated segments clearly at the end of the introduction.

Methods

Page 12, lines 130-131, researchers referred readers to their previous published paper, which may not be accessible.

It is suggested to mention the reason of attrition clearly.

Since researchers aimed to assess criterion validity of the scales, I’m wondering which scale was used as the criterion in the study. Its psychometric characteristics including validity as well as reliability should be described.

Reviewer #2: The study from a scientific point of view seems to be well done and presents good results, from where to derive valid results. The problem discussed is currently valid. The summary sufficiently informs about the content of the paper. I have minor comments:

-The questionnaire was well explained and internal validity was reported, what about validity?

-The final conclusion of study was missed.

-Some of references were before 2000, please update them.

6. PLOS authors have the option to publish the peer review history of their article (what does this mean?). If published, this will include your full peer review and any attached files.

Reviewer #1: Yes: Maryam Rassouli

Reviewer #2: Yes: Leila Jahangiry

---

## [Author Response · Author response to Decision Letter 0]

3 Mar 2020

Reviewer #1

1. Background on depression in children. Good point which we had omitted. We have now added that prevalence is considerably lower and not increasing in children while we also added that the degree of clinical recognition in children is not known. 

2. Role of the K-SADS-PL in the introduction section. We have specified the different criterion test that arrived at opposing cut offs (same for boys or girls versus a lot higher cut off for girls). The criterion was a clinician diagnosis utilising K-SADS-PL in one study while the criterion was just a cut-off on a self report in the other study. Thus, the role of K-SADS as a more reliable diagnostic test than a self report form is explained. 

3. Aims of study as clear separate segments. The aims are now separated by a line break and are specified as just two and more clearly phrased as ”to …”

4. To describe the reason for attrition more clearly. A more detailed description of the inclusion process are included as well as a specification of the missing 35 parental MFQ was due to administrative problems.

5. Criterion scale in the study. Very important point and possible ”reference test” was not a clear description of the role of LEAD in this major design issue. We have now clearly stated in several segments (aims, statistics, results…heading of table 2 and discussion) that a LEAD diagnosis was the criterion in the study. However, it is not possible to describe the validity of the LEAD while the interrater reliability of the LEAD is now described in the methods section.

Reviewer #2

1. Validity of the MFQ and SMFQ in the methods section. As the validity of the scale is the main focus of the paper, we chose to describe and discuss this topic in the introduction. This discussion is the base for the summing up of the background and aims of the study. Therefor, we would still prefer to describe previous research on validity of MFQ in the introduction. 

2. A final conclusion (same as in abstract) is added. Thanks for this suggestion!

3. References before 2000. We would like to keep these references as the original studies on MFQ were published in the late 90´s. However, we are refering to just about every relevant study MFQ and SMFQ including all with more recent dates.

---

## [Editor Report · Decision Letter 1]

5 Mar 2020

Screening efficiency of the Mood and Feelings Questionnaire (MFQ) and Short Mood and Feelings Questionnaire (SMFQ) in Swedish help seeking outpatients

PONE-D-19-26460R1

Dear Dr. Jarbin,

We are pleased to inform you that your manuscript has been judged scientifically suitable for publication and will be formally accepted for publication once it complies with all outstanding technical requirements.

With kind regards,

Ali Montazeri

Academic Editor

PLOS ONE
---

## [Editor Report · Acceptance letter]

9 Mar 2020

PONE-D-19-26460R1 

Screening efficiency of the Mood and Feelings Questionnaire (MFQ) and Short Mood and Feelings Questionnaire (SMFQ) in Swedish help seeking outpatients 

Dear Dr. Jarbin:

I am pleased to inform you that your manuscript has been deemed suitable for publication in PLOS ONE. Congratulations! Your manuscript is now with our production department. 

With kind regards,

on behalf of

Professor Ali Montazeri 

Academic Editor

PLOS ONE